# Non-Cryoprecipitation Separation of Coagulation FVIII and Prothrombin Complex Proteins by Pseudoaffinity Calcium Elution Chromatography Using Anion Exchange Resin

**DOI:** 10.3390/ph15101192

**Published:** 2022-09-27

**Authors:** Gabriel Pinna Feliciano, Sara Hayama Arimori, Vinicius Watanabe Nakao, Joice Rodrigues Dos Santos, Elizabeth A. L. Martins, Marcelo Porto Bemquerer, Elisabeth Cheng

**Affiliations:** 1Laboratório de Desenvolvimento de Processos, Instituto Butantan, São Paulo 05503-900, SP, Brazil; 2Programa de Pós-Graduação Interunidades em Biotecnologia, Universidade de São Paulo, São Paulo 05508-900, SP, Brazil; 3Laboratório de Biológicos Recombinantes, Instituto Butantan, São Paulo 05503-900, SP, Brazil; 4Embrapa Gado de Leite, Juiz de Fora 36038-330, MG, Brazil

**Keywords:** pseudoaffinity chromatography, non-cryoprecipitation, FVIII, prothrombin complex, anion exchange chromatography

## Abstract

Hemophilia A is treated with human plasma coagulation factor VIII (FVIII) replacement therapy and Hemophilia B with coagulation factor IX, which is purified from prothrombin complex concentrate (PCC). In this paper we evaluated the separation of FVIII and PCC by directly loading raw thawed plasma to an anion exchange resin (AEX). Under this relatively high ionic strength, most of the plasma proteins such as albumin, immunoglobulins and others were not adsorbed. Five resins commonly used in protein purification (plasma fractionation) were tested. With all resins, PCC was eluted by pseudoaffinity in a calcium gradient step. Afterwards, FVIII could be recovered with a good yield and high purification factor in the salt gradient step with 400–500 mM NaCl. Using ANX Sepharose FF and Q Sepharose FF, the CaCl_2_ elution step was introduced after the intermediate wash with 200 mM NaCl, whereas using DEAE Sepharose FF, Fractogel EMD TMAE and Fractogel EMD DEAD, PCC eluted after the wash of the unbound proteins. Our results indicate that three important fractions: (1) albumin, immunoglobulin etc.; (2) PCC; and (3) FVIII can be separated in one chromatographic AEX column and the delicate and troublesome cryoprecipitation can be eliminated, making the purification of blood products faster and cheaper.

## 1. Introduction

Hemophilia is an inherited bleeding disorder in which the hemostatic defect results from deficiency of coagulation factor VIII (FVIII) in hemophilia A or factor IX (FIX) in hemophilia B. Traditional treatments for hemophilia have largely worked by directly replacing the missing coagulation factor [1].

FVIII concentrates are used in the treatment of Hemophilia A [2]. This coagulation factor is an essential glycoprotein for the clotting process. It circulates in the bloodstream complexed with von Willebrand factor, which acts as a transporter [3] and prevents the degradation of FVIII by the action of serine proteases, including activated Protein C and activated FX [4]. FVIII is a high molecular mass (300 kDa) and labile protein present in very low concentrations in plasma (150 ng mL^−1^) [4,5].

Coagulation factor IX (FIX) belongs to the family of vitamin K-dependent proteins [6] along with coagulation factors II (FII or prothrombin), VII (FVII), and X (FX), which have procoagulant activity, and Protein C (PC), Protein S (PS) and Protein Z (PZ), which have anticoagulant activity. Vitamin K-dependent proteins have molecular masses between 50 kDa and 72 kDa [7]. At the N-terminus, they have a region with 9 to 12 γ-carboxyglutamic acid residues, called the GLA domain, that contain Ca^2+^ sites. The binding of Ca^2+^ ions with the carboxyls of γ-glutamic acids exposes the side chains of hydrophobic residues, favoring the interaction with the phospholipids present on the surface of the membranes where the catalytic complexes are formed which lead to the sequential activation of the coagulation [6].

Currently, almost all blood products purification plants use the modified Cohn method [8], which consists of successive precipitations with increasing ethanol concentration, in different pH and in a cold chamber, followed by chromatographic steps used to obtain products with a high degree of purity. In a typical fractionation scheme, plasma bags are thawed at 1 to 4 °C and cryoprecipitate is isolated by refrigerated centrifugation, providing concentrates of FVIII, von Willebrand factor and fibrinogen [8]. The capture of FVIII is processed by anion exchange chromatography. The vitamin K dependent proteins remain in the supernatant (called cryo-poor plasma), along with albumin, IgG and other proteins. They are frequently captured by anion exchange chromatography, resulting in a mixture of intermediate purity, called the prothrombin complex concentrate (PCC) [9,10]. Highly purified FIX, used in hemophilia B replacement therapy, is isolated from PCC [9,11].

Anion exchange resins are widely used in the plasma fractionation processes [3,12,13,14,15] due to their high capacity, good yields and low cost compared to other chromatographic techniques. Upon loading whole plasma directly onto an anion exchange column, without the cryoprecipitation step, FVIII and PCC are strongly adsorbed and cannot be eluted separately using NaCl gradient [16,17]. A second column was needed to separate FVIII from PCC proteins, and it was successfully achieved by immobilized metal affinity chromatography [16,18]. Conventional affinity chromatography involves the immobilization of a ligand for which the protein of interest has an affinity. Pseudoaffinity elution in ion exchange chromatography differs from affinity chromatography in that there is no ligand immobilized on the matrix for which the protein of interest has affinity, and this matrix can be a conventional ion exchange resin or hydrophobic interaction [19]. When binding to the Ca^2+^ ion, the vitamin K-dependent proteins undergo a conformational change which will lead to a decrease in the exposed negative charges, modifying their affinity for the resin, disconnecting them from the column with lower ionic strength. Based on this concept, the purification of recombinant coagulation factors rFII, rFIX and inhibitors rPC and rPS from their culture media has been described in the literature [19,20,21].

To eliminate the cryoprecipitation step and the second chromatography, in this paper we investigated the selective elution of FVIII and PCC proteins from anion exchange columns, taking advantage of the conformational change of prothrombin complex proteins in the presence of low concentrations of Ca^2+^ ions. We evaluated the elution of PCC by changing the Ca^2+^ concentration in the buffer solutions and then FVIII was eluted by high concentrations of NaCl. Five conventional resins and three buffer solutions were tested. 

## 2. Results

### 2.1. Purification on Q Sepharose FF–Ca^2+^ 5 mM to 100 mM Linear Gradient and 15 mM CaCl_2_ and 65 mM CaCl_2_ Stepwise Gradient

A linear 5 mM to 100 mM Ca^2+^ gradient was carried out after the intermediate wash with Buffer B (citrate 25 mM, NaCl 200 mM, CaCl_2_ 5 mM, pH 6.0). The chromatogram of the purification (Figure 1a) shows that two peaks eluted with the increase of Ca^2+^ and no peak was observed when NaCl was increased to 500 mM, using Buffer C (citrate 25 mM, NaCl 500 mM, CaCl_2_ 5 mM, pH 6.0). The activity profile shown in Figure 2a indicates that FIX eluted first, between F3 and F6 fractions, that is, between, 5.4 and 17.9 mM CaCl_2_, while FVIII eluted from F6 to F13, that is, between 17.9 and 63.6 mM CaCl_2_. The protein profile of the collected fractions is shown in the SDS-PAGE gel (Figure 3a). Based on this result, we evaluated a protocol employing a two stepwise gradient with 15 and 65 mM CaCl_2_ after the intermediate wash 200A with Buffer B (Figure 4a,b). Under this condition, FIX was successfully recovered separately from FVIII. FIX was recovered with 86% activity with Buffer B + 15 mM CaCl_2_. FVIII activity was recovered in the same buffer containing CaCl_2_ 65 mM fraction (Figure 4b, lane “65A”) with 60% activity and a purification factor of 201 (Table 1).

### 2.2. Purification on ANX Sepharose FF–Ca^2+^ 5 mM to 25 mM Linear Gradient and Citrate Buffer

The next resin tested was ANX Sepharose FF. Since FIX eluted very well on Q Sepharose FF using CaCl_2_ 15 mM, in this experiment a linear calcium gradient was carried out from 5 to 25 mM. The obtained chromatogram is shown in Figure 1c. Two peaks were observed with the increase of calcium concentration. Recovery of FIX and FII is shown in Figure 2c. FII and FIX are in the second peak. FIX desorbed between F6 and F9, that is, 10.3 mM and 16.7 mM CaCl_2_, before prothrombin, which desorbed from the column between F7 and F13, that is, between 11.9 mM and 21.4 mM CaCl_2_, respectively. FVIII was recovered mainly with buffer C with 51% yield and purification factor 157. Analysis by SDS-PAGE is presented in Figure 3c. The higher molecular mass proteins observed on the Q Sepharose FF gel (Figure 3a), due to the higher CaCl_2_ concentration used (65 mM), were not observed on ANX Sepharose FF purification gel where the highest CaCl_2_ concentration achieved was 25 mM.

### 2.3. Purification on DEAE Sepharose FF–Ca^2+^ 5 mM to 50 mM Linear Gradient 

During the purification of plasma on DEAE Sepharose FF using the protocol without the CaCl_2_ gradient step, 35% of FVIII eluted in the 200 mM intermediate wash. Therefore, the calcium gradient was introduced after the wash of the unbound proteins.

The chromatogram of the purification is shown in Figure 1d. The maximum elution of FIX and prothrombin were obtained with CaCl_2_ 18.8 mM (F8) and 29.8 mM (F10), respectively (Figure 2d). With DEAE Sepharose FF, like the purification using Q and ANX Sepharose FF, FIX also desorbed before prothrombin. FVIII eluted mainly with NaCl 400 mM (Buffer D) with 65% yield (Figure 2d) and a purification factor of 125 (Table 1). In this fraction, no prothrombin activity was found, and the FIX activity found was negligible (0.1%), indicating that the separation of FVIII from the prothrombin complex proteins succeeded. Since the concentration of NaCl present in the citrate buffer was 85 mM, the concentration of CaCl_2_ required to elute the vitamin K dependent proteins was higher than the concentrations of CaCl_2_ of the purifications on Q and ANX Sepharose FF columns, where the CaCl_2_ gradient step was carried out with the citrate buffer containing NaCl 200 mM. Therefore, the good separation of vitamin K dependent proteins and FVIII depends on the combination of the NaCl e CaCl_2_ concentrations. SDS-PAGE analysis is presented in Figure 3d.

### 2.4. Purification on Fractogel EMD TMAE–Ca^2+^ 5 mM to 50 mM Linear Gradient 

In the purification on Fractogel EMD TMAE without CaCl_2_ gradient, employing the protocol with stepwise NaCl gradient at 200 mM and 500 mM (in buffers B and C), 37% of the FVIII activity was recovered in the Buffer B and 63% in the Buffer C fractions, while FIX eluted in the NaCl 500 mM with 48% yield. Therefore, for the elution of vitamin K dependent proteins, a calcium gradient was introduced after the wash of the unbound proteins.

Figure 2b shows that in the purification on Fractogel EMD TMAE, FIX and prothrombin eluted in the same fractions starting from F4, the concentration of Ca^2+^ was 13.7 mM and the maximum elution was observed with F6 and F7, where Ca^2+^ was 21.8 and 26.8 mM, respectively. The activity of FVIII was found mainly with the elution of Buffer D (25 mM sodium citrate, NaCl 400 mM and 5 mM CaCl_2_, pH 6.0), which gave a 51% yield and a purification factor of 142 (Table 1).

### 2.5. Purification on Fractogel EMD DEAE–Ca^2+^ 5 mM to 50 mM Linear Gradient 

The calcium gradient was carried out after the wash of the unbound proteins. A chromatogram of the purification is shown in Figure 1e. In the purification on Fractogel EMD DEAE employing a linear 5 mM to 50 mM Ca^2+^ gradient 73% of FVIII eluted with buffer D and purification factor of 132 (Table 1). FIX and Prothrombin elutted in the same fractions starting from F5, where the concentration of Ca^2+^ was 15.7 mM. Maximum elution was observed with F8–F10, where Ca^2+^ was between 31.0 mM and 42.0 mM (Figure 2e). These results indicate that the vitamin K dependent coagulation factors bind stronger on Fractogel EMD DEAE than on TMAE. An SDS- gel analysis of the recovered fractions is shown in Figure 3b.

Our results indicate that prothrombin complex could be eluted with CaCl_2_ without affecting FVIII activity using all five tested resins.

### 2.6. Purification on ANX Sepharose FF–Calcium 5 mM to 25 mM Linear Gradient and Bis-tris Buffer 

The chromatogram of the purification of plasma on ANX Sepharose FF employing a linear 5 to 25 mM Ca^2+^ gradient and Bis-tris buffer presented in Figure 1f shows that one peak eluted with the increase of calcium concentration, in contrast to the purification using citrate buffer, where two peaks were observed during elution with calcium (Figure 1c). It is interesting to note that the peak eluted Buffer B (200A), observed using the citrate buffer (Figure 1b), was almost absent in this purification. FVIII was recovered in the NaCl 500 mM fraction (Figure 2f) with 67% yield and purification factor of 211 (Table 1). No FVIII activity was found in any other fraction (Figure 2f). On the other hand, FIX activity was observed in all fractions of the calcium gradient, and in the NaCl 500 mM fraction (26%) along with FVIII (Figure 2f).

### 2.7. Purification on ANX Sepharose FF–Ca^2+^ 5 mM to 25 mM Linear Gradient and MES Buffer 

In the purification using MES buffer, the chromatogram (Figure 1g) as well as the FVIII and FIX activity recovery profiles (Figure 2g) were very similar to the observed with the purification using Bis-tris buffer (Figure 1f and Figure 2f). FVIII was recovered with 52.9% yield (Figure 2g) and a purification factor of 125 (Table 1). FIX was mainly recovered in the calcium gradient fractions, but 27% of the activity was recovered in the NaCl 500 mM, along with FVIII. Therefore, the MES buffer gave similar results to those observed with Bis-tris.

### 2.8. Stepwise CaCl_2_ Gradient in the Purification on ANX Sepharose 

After verifying that the vitamin dependent proteins could be well separated from FVIII using a linear gradient, we created a protocol containing a CaCl_2_ stepwise gradient. For that, ANX Sepharose FF resin was chosen.

Two stepwise calcium gradient purifications were tested: with one step of 25 mM CaCl_2_ and with two steps of 10 mM CaCl_2_ and 25 mM CaCl_2_.

The chromatogram of the purification with one calcium gradient step with 25 mM CaCl_2_ is presented in Figure 4b. Two peaks in the calcium gradient were observed. The yield of FVIII eluted with buffer C was 67% with a purification factor of 250 (Table 1). In this experiment it was possible to determine the activity of Protein C, since the activity in the fractions collected in the CaCl_2_ linear gradient were lower than the detection limit of the chromogenic method. The yield of Protein C in this experiment was 44% and the purification factor was 126.

The chromatogram of the purification with calcium gradient step with CaCl_2_ 10 mM and 25 mM is shown in Figure 4c. In the insert of Figure 4c, it is possible to see that the two peaks were well resolved. The coagulation proteins Prothrombin, FIX and Protein C were recovered in the 25 mM CaCl_2_ fraction. The recovery of Prothrombin and FIX was 100%, and the purification factors were 689 and 496, respectively. The yield of Protein C was 56% and the purification factor was 162. The yield of FVIII in the fraction “500” was 59%, with a purification factor of 220 (Table 1).

### 2.9. FVIII and FIX Dynamic Binding Capacity and NaCl Elution in Plasma Purification on ANX Sepharose FF

The loading profiles of FVIII and FIX in the purification of plasma on ANX are quite different (Figure 5). While the FVIII starts leaking from the column after the loading of 10 CV of plasma, the FIX activity measured after loading 20 CV of plasma was negligible. This result indicates that using plasma as starting material, the binding capacity of FIX is higher than the binding capacity of FVIII. 

The FVIII elution peak is very wide. Activity of FVIII was observed in all collected fractions, the maximum being roughly the same as that observed for FIX in F8, where the NaCl concentration was 440 mM and conductivity 45 mS cm^−1^ (Figure 6). FIX has a sharp peak between F5 and F12, that is, between 320 mM NaCl and 600 mM NaCl, respectively.

### 2.10. Analysis by Mass Spectrometry of the Fractions Collected in the Calcium Gradient of the Purifications on ANX Sepharose FF

During the study using different resins, we observed, in the non-reducing SDS-PAGE, that non-vitamin dependent proteins coeluted with the vitamin K dependent proteins, although their conformations are not expected to be affected by the variation of the calcium concentration. Figure 7 shows the non-reducing gel of the samples presented in Figure 3c. It was possible to observe high molecular mass proteins present in all collected fractions (Figure 7).

To identify the proteins that eluted with the increase of calcium concentration on ANX Sepharose FF purifications, we analyzed the bands of the SDS PAGE reducing gel of six different fractions eluted by a 5 to 25 mM calcium gradient using mass spectrometry. We analyzed the bands marked with arrows (Figure 8). Spots from fractions F2, F9, F12 and F15, which were taken from the reducing SDS-PAGE (7.5% acrylamide,) are shown in Figure 8a, which is the same gel shown in Figure 3c. Spots from fractions F4 and F9 were taken from 5 to 15% acrylamide reducing SDS-PAGE (Figure 8b). The results are shown in Table 2. The identified proteins were Complement C4 binding protein, Fibrinogen, Complement C4 and Prothrombin.

We also analyzed spots of lane “25A” of the SDS-PAGE reducing gel shown in Figure 4d from the purification on ANX Sepharose FF using a stepwise 25 mM CaCl_2_ gradient. The spots analyzed are indicated by arrows in Figure 9a. The same four proteins were identified: Complement C4 binding protein, Fibrinogen, Complement C4 and Prothrombin (Table 3).

Finally, spots of the fraction that elutes with 25 mM CaCl_2_ from purification using two steps CaCl_2_ gradient (Figure 4f, lane “25”) were analyzed. Nine lanes of the gel were loaded with the fraction “25” in a non-reducing polyacrylamide gel 6% (Figure 9b). Bands shown in Figure 9b were excised horizontally and treated with trypsin, DTT and iodoacetamide according to the described method. Part of the prepared samples were directly analyzed by mass spectrometry and part was analyzed by RF-HPLC-C18 and then by mass spectrometry. In this experiment, two more vitamin K dependent proteins were identified, FIX and Protein Z (Table 4). This last protein could only be identified by the mass spectrometry of a compound eluted from the RP-HPLC C-18 analysis of the lowest molecular mass bands of the non-reducing gel shown in Figure 9b.

Prothrombin (FII) was identified in all preparations because this protein is in greater abundance. While the concentration of FII in plasma is 90 µg mL^−1^, the second most abundant vitamin K dependent protein is Protein S, with 30 µg mL^−1^ and is complexed with the C4b binding protein. The other proteins of the family were found in a concentration more than ten times lower. Only with the gathering of proteins from the bands of non-reducing gels was it possible to identify FIX. Factor IX has a plasma concentration of 3 to 5 µg mL^−1^ [22] and a lower molecular mass (56 kDa) than Prothrombin (72 kDa). The analysis of one of the fractions obtained in the HPLC analysis of sample 9 made it possible to identify a fragment of Protein Z, which is a vitamin K-dependent factor of 62 kDa and has a plasma concentration of 1.2 to 2.9 µg mL^−1^. It is very likely that other not identified vitamin K dependent factors are also present. 

Of the non-vitamin K dependent proteins identified, two are complement system proteins. C4bBP is a high molecular mass protein (570 kDa), which, in addition to the complement system, participates in coagulation through complex formation with Protein S and is present in high concentrations in plasma (150 µg mL^−1^). Protein C4 is also a high molecular mass protein (202 kDa) present in high concentrations in plasma (600 µg/mL). Fibrinogen is an important molecule in hemostasis and tissue repair. It is also a molecule of high molecular mass (340 kDa), present in high concentrations in plasma (3 g L^−1^). The presence of these proteins with similar concentrations along the entire gradient (Figure 7) suggests that these proteins did not elute due to the increase in CaCl_2_ concentration. The high concentration and the low diffusion rate through the resin due to the size could explain the presence of these proteins together with the vitamin K-dependent proteins. 

## 3. Discussion

The chromatograms shown in Figure 1 are very similar and simple, but the elution profile of FVIII and FIX, and therefore PCC, differ significantly. We tested two strong anion exchange resins (Q Sepharose FF and Fractogel EMD TMAE) and three weak anion exchange resins (ANX and DEAE Sepharose FF, Fractogel EMD DEAE). The matrix of Sepharose resins is agarose and the matrix of the Fractogel resins is metacrylate. No general protocol could be established based on the strength of the anion exchanger nor the chemical nature of the matrix. 

In the purifications on Q and ANX resins, the CaCl_2_ gradient step following the 200A wash worked very well. PCC (FIX) and FVIII have more affinity to ANX Sepharose FF resin than Q Sepharose FF. FIX started the desorption from the Q column with 5.4 mM CaCl_2_ (Figure 2a) and from the ANX column with 10.3 mM CaCl_2_ (Figure 2c). FVIII started the desorption from Q Sepharose FF with 17.9 mM CaCl_2_ (Figure 2a) and did not elute from the ANX Sepharose FF column with 25 mM CaCl_2_ (Figure 2c). With both columns it was possible to establish a stepwise protocol to elute FVIII and PCC (FIX) separately. Noteworthy is that it was possible to elute FVIII from the Q Sepharose FF column by increasing CaCl_2_ concentration to 65 mM from 25 mM citrate buffer, pH 6.0 containing NaCl 200 mM without formation of coagulum (Figure 4a, lane 65A). However, the FVIII yield obtained in this purification was comparable to that obtained with the FVIII eluted with 500 mM NaCl [20] and the purification factor was comparable to that observed with all other resins tested in this work employing the CaCl_2_ gradient step (Table 1). The absence of the peak 500 in Figure 4a and the similarity of the protein profile lane 65A of Figure 4a (purification on Q resin) and lane “500” of Figure 4d,f (purification on ANX) indicate that FVIII eluted due to increase of the ionic strength. No improvement was observed by eluting FVIII by increasing CaCl_2_ concentration. Furthermore, work with highly concentrated Ca^2+^ solutions is not recommended because of the known property of Ca^2+^ ions to promote coagulation. For the purifications on the Fractogel EMD, TMAE, and DEAE resins, the CaCl_2_ gradient step was carried out immediately after the reequilibration wash of the unbound proteins. The main reason for introducing the calcium gradient step with lower ionic strength was the loss of FVIII. Therefore, much higher concentrations of CaCl_2_ were required to elute PCC(FIX) (Figure 2b,c,e). In the purification of FVIII from cryoprecipitate carried out by Mori et al. [3], the yield and purity of this coagulation factor was much higher using TMAE than DEAE. In contrast, our results showed that DEAE gave a higher yield and both resins resulted in FVIII with similar purity (Table 1). The purification profile on DEAE Sepharose FF shared common features with Sepharose FF resins and with the Fractogel resins. FIX started eluting before Prothombin as on the other Sepharose resin purifications. As with the Fractogel resins, the Ca^2+^ gradient step followed the wash of the unbound proteins and FVIII eluted with NaCl 400 mM in a citrate buffer (buffer D). The SDS-PAGE analyses presented in Figure 3 shows that the protein profile is slightly different in all purifications, possibly due to the differences in selectivity of each resin used. The results presented in this work showed that the proposed purification method could be successfully applied to all five resins tested, provided that the CaCl_2_ and NaCl concentrations in citrate buffer are carefully determined. 

Interestingly, the citrate buffer played a significant role in the purification profile. The citrate buffer is used in FVIII purifications because of its anticoagulant activity, but it is also able to chelate Ca^2+^ ions. For this reason, purifications on ANX employing two other buffers were tested. Bis-tris and MES buffers were chosen for comparison because they do not chelate calcium ions (or very weakly), have a pKa near 6 (6,46 and 6,15, respectively), and have completely different chemical structures (amine and sulfonic acid, respectively). Compared with the citrate buffer, using Bis-tris and MES buffers, the same 200A peak was not observed (Figure 1c,f,g), and separation of FVIII and FIX was not successful, because 20 to 30% of PCC (FIX) did not elute with CaCl_2_ 25 mM (Figure 2c,f,g). In contrast to what we thought, our results suggest that a higher concentration of CaCl_2_ would be needed to elute all FIX. Somehow the citrate buffer that interacts with the calcium ions favored the elution of FIX. Purifications of recombinant Protein C by Mono Q and Q Sepharose resins were successfully carried out using a 20 mM Tri-HCl buffer with a pH of 7.4 containing NaCl 150 mM and 10 mM CaCl_2_, most likely because the PCC (FIX) affinity for the Q resin is weaker than for the ANX resin [23].

The yield and the purification factor of FVIII in each experiment are shown in Table 1. In the purification of plasma on ANX Sepharose FF without the CaCl_2_ gradient step, the FVIII recovered activity in fraction “500” described by Verinaud et al. [16] was 69 ± 18. The results obtained in this work are in line with this result. On the other hand, the purification factors obtained in this work, regardless the resin used, were much higher than those described in the work of Verinaud et al. [16] (110 ± 11). These results indicate that the CaCl_2_ gradient step did not have a negative affect on FVIII activity and significantly improved the purification factor.

Aiming at scaling up the purification, two stepwise CaCl_2_ gradient protocols using ANX and the citrate buffer were evaluated: one with one step of 25 mM CaCl_2_ and one with two steps of 10 and 25 mM CaCl_2_. The obtained results indicate that PCC (FIX) eluted from the 25 mM CaCl_2_ and was successfully separated from FVIII, but the additional washing step with 10 mM CaCl_2_ did not lead to improvement either in the yield or in the purification factor. FIX and Prothrombin, components of PCC, were recovered with high yield and purification factor. Protein C, on the other hand, was recovered with a lower yield (44% and 56%) and purification factor (126 and 162). These results are in accordance with the result obtained with the elution of this protein by 500 mM NaCl [16]. In contrast, recombinant Protein C could be recovered with high yield (85%) (and 96% purity) from culture media on Q Sepharose FF using the Ca^2+^ gradient [23]. 

The binding capacity of 10 CV of FVIII is highly favored by the large pore size of the resin (Figure 5). We have previously observed that for the purification of FVIII, without the Ca^2+^ gradient, ANX gave a higher yield (69 ± 18%) and purification factor (110 ± 11) [16] than Q, which gave a 48 ± 19% yield and a purification factor of 55 [17]. ANX resin has a lower agarose content (4%) than Q resin (6%), and therefore a larger pore size, which favors the purification of high molecular mass proteins, like FVIII, which has 300 kDa. However, the results obtained using these two resins, with a Ca^2+^ gradient, were similar (Table 1). Figure 5 shows that FIX starts eluting from the column in F6, in which the NaCl concentration is about 340 mM with a conductivity of about 33 mS cm^−1^ (Figure 6). To be completely eluted it requires NaCl 450 mM to NaCl 500 mM, which is near 50 mS cm^−1^. Using the Ca^2+^ gradient, FIX is completely eluted with a buffer of 25 mM citrate, NaCl 200 mM, CaCl_2_ 25 mM, and a pH of 6.0, in which the conductivity is 26 mS cm^−1^ (Figure 4b,c). It is therefore clear that by increasing the concentration of CaCl_2_ to 25 mM, the protein is displaced from the column by pseudoaffinity.

Figure 9 shows that the 3 non-vitamin K dependent proteins (C4bBP, C4 and Fibrinogen), identified by mass spectrometry, are the contaminant proteins present in the obtained PCC. It seems that these proteins are “trapped” inside the ANX Sepharose FF resin beads. Considering the SDS-PAGE analyses shown in Figure 3, bands of these proteins are present in other, if not all, gels. It seems that these proteins are present in the collected fractions of all resins and in the FVIII containing fractions. This observation will have implications for future purifications.

Figure 1 shows the chromatograms of the purifications with all investigated resins. They gave one short and wide peak in the calcium gradient step, except for the purification on ANX Sepharose FF, which presented two peaks. The first peak eluted with CaCl_2_ 10 mM and the second peak, which contained the vitamin K dependent proteins, eluted with 25 mM CaCl_2_ (Figure 4f). The same bands that eluted with 10 mM CaCl_2_ are present in the purifications by all tested resins, but only in the purification by ANX Sepharose FF part of these proteins eluted separately. One possible explanation is that a small part of the high molecular mass proteins, identified by mass spectrometry and were observed in all lanes of Figure 7, was washed from the ANX Sepharose FF because of the large pore size and lower rigidity of this resin. Therefore, only the vitamin K-dependent proteins eluted with the increasing Ca^2+^ concentration. 

We identified in the Ca^2+^ elution fraction four out of 7 PCC proteins: prothrombin, FIX and Protein C were identified by the biochemical chromogenic method, and prothrombin, FIX and Protein S were identified by mass spectrometry. Because of the similarity in their structure, it is likely that other PCC proteins can be eluted by pseudoaffinity. Therefore, we showed in this work that the purification by pseudoaffinity described for PCC recombinant proteins [19,20,21] can be successfully applied to the purification of plasma PCC. 

Compared with the traditional Cohn method, in the proposed purification method it is possible to obtain an enriched FVIII fraction without the need of cryoprecipitation, thus avoiding refrigerated thawing and centrifugation at low temperatures. Attempting to avoid cryoprecipitation is not new [24,25,26,27]; however it was necessary to use 2 ion exchange columns to separate FVIII from PCC, one of them being used in batch mode, which makes it more difficult to scale up the process. Finally, the method we describe in this work can be easily integrated into any fractionation process. The fraction of the non-adsorbed proteins, which can be called the clotting factor-poor fraction, containing roughly 99% of the plasma proteins including albumin, immunoglobulins and other proteins, can continue in the plasma fractionation process. The elimination of 2 process steps as well as the need of controlled thawing and refrigerated centrifugation of the cryoprecipitate can contribute significantly to decrease the process time and cost.

## 4. Materials and Methods

Fresh frozen plasma (FFP) bags were kindly provided by the Ribeirão Preto Hemocenter and the Blood Collection Beneficent Association (Colsan) of São Paulo, Brazil. The anion exchange resins Q, ANX and DEAE Sepharose FF resins were purchased from Cytiva (Uppsala, Sweden). The anion exchange resins Fractogel EMD, TMAE, and DEAE were kindly provided by Merck (Darmstadt, Germany). A FVIII chromogenic assay kit Coatest^®^ SP4 FVIII and Protein C assay kit COAMATIC^®^ Protein C were purchased from Chromomogenix (Bedford, MA, USA). FIX (Biophen Factor IX) and Prothrombin (Biophen Prothrombin) chromogenic assay kits were purchased from Hyphen Biomed (Neuville sur Oise, France). All reagents were of analytical grade. All solutions were prepared with water from the purification system Millipore RiOs. All chromatographies were carried out using Äkta^®^ Explorer or Purifier systems run with Unicorn 5.01 software (Cytiva). All resins were packed in XK26/20 columns (Cytiva).

### 4.1. Chromatography General Procedure 

To establish the more suitable purification step for the calcium gradient, we first considered the protocol that we used for the purification of FVIII on the Q Sepharose FF [17] and on the ANX Sepharose FF [16] columns using an NaCl stepwise gradient: 200 mM and 500 mM. We then considered the protocol described by Yan et al. [23], in which the recombinant vitamin K dependent PC eluted from an anion exchange column using Tris-HCl buffer, pH 7.4 containing 150 and 10 mM CaCl_2_. Therefore, in the first protocol tested, calcium gradient, used to elute vitamin K dependent proteins, was introduced between the NaCl 200 mM and 500 mM steps.

In all experiments, five bags of FFP were thawed in a water bath at 37 °C, and the pH of the pool was adjusted to pH 6.0 with 0.2 M citric acid and directly applied to resins. Columns were equilibrated with five column volumes (CV) of Buffer A (25 mM sodium citrate, 85 mM NaCl and 5 mM CaCl_2_, pH 6.0). Five CV of the pool of FFP were applied to the columns and the unbound proteins were washed with 10 CV of Buffer A. The fraction containing the unbound proteins was called flow through (FT). After the wash of the unbound proteins, the procedure followed in each experiment is described in the corresponding item.

### 4.2. Chromatography of Plasma on Q Sepharose FF

The flow rate was 15 mL min^−1^ during plasma application and 25 mL min^−1^ during other purification steps. The volume of the Q Sepharose FF column was 46 mL.

#### 4.2.1. Q Sepharose FF Chromatography with Ca^2+^ 5 mM to 100 mM Linear Gradient

Equilibration, sample, and washings with 10 CV of Buffer A, according to Section 4.1 and Section 4.2, and 10 CV of Buffer B (25 mM sodium citrate, 200 mM NaCl and 5 mM CaCl_2_, pH 6.0) (200A) were carried out. Next, 10 CV of 5 mM to 100 mM CaCl_2_ (in Buffer B) linear gradient and 5 CV stepwise Buffer B + 100 mM CaCl_2_ were undertaken. After a second wash with 5 CV of Buffer B (200B), elution with 4 CV of Buffer C (25 mM sodium citrate, 500 mM NaCl and 5 mM CaCl_2_, pH 6.0) was performed.

Resin was regenerated by sequentially washing with 2 CV 1M NaOH (with 1 h pause), 5 CV 2M NaCl and 5 CV purified water and stored in 0.2 M NaOH.

#### 4.2.2. Q Sepharose FF Chromatography with Ca^2+^ 15 mM and 65 mM Stepwise Gradient

Equilibration, sample, and washings with Buffers A and B were carried out as indicated in Section 4.1, Section 4.2 and Section 4.2.1 Protein elution with 5 CV Buffer B + 15 mM CaCl_2_, and 5 CV Buffer B + 65 mM CaCl_2_ were performed. Washing with Buffer B (200B) and elution with Buffer C were then carried out according to Section 4.2.1. Resin regeneration and storage were carried out as in Section 4.2.1.

### 4.3. Chromatography of Plasma on ANX Sepharose FF

The volume of ANX Sepharose FF was 48 mL, and the used flow rates were as indicated in Section 4.2.

#### 4.3.1. ANX Sepharose FF Chromatography with Ca^2+^ 5 mM to 25 mM Linear Gradient with Citrate Buffer

Equilibration, sample, and washings with Buffers A and B were carried out as indicated in Section 4.1, Section 4.2.1, and Section 4.3. Next, 10 CV of linear gradient 5 mM to 25 mM CaCl_2_ (in Buffer B) and 5 CV stepwise Buffer B + 25 mM CaCl_2_ were carried out. They were then washed with Buffer B (200B) and the elution with Buffer C was performed as in Section 4.2.1. 

Resin was regenerated by sequentially washing with 2 CV 1M NaOH (with 1 h pause), 5 CV 2M NaCl and 5 CV purified water and stored in 10 mM NaOH.

#### 4.3.2. ANX Sepharose FF Chromatography with Ca^2+^ 5 mM to 25 mM Linear Gradient with Bis-tris or MES Buffers

Because the citrate buffer interacts with Ca^2+^ ions, we carried out purifications employing Bis(2-hydroxyethyl) amino-tris(hydroxymethyl)methane (Bis-tris) and 2-(N-morpholino)-ethane sulfonic acid (MES) buffers, which have pKa 6.46 and 6.15, respectively, and were suitable for purifications at pH 6.0 and do not chelate divalent ions.

In these experiments, instead of the citrate buffer, Bis-Tris or MES buffers were used, except in the elution of FVIII, which was carried out with citrate (Buffer C). All other experimental conditions followed the procedure described in item Section 4.3.1.

#### 4.3.3. ANX Sepharose FF Chromatography with Ca^2+^ Stepwise Elution 

Equilibration, sample, and washings with Buffers A and B were carried out as indicated in Section 4.1, Section 4.2.1. and Section 4.3. After 200A washing, elutions with 5 CV Buffer B + 25 mM CaCl_2_ (protocol with one Ca^2+^ step) or 5 CV Buffer B + 10 mM CaCl_2_, and 5 CV Buffer B + 25 mM CaCl_2_ (protocol with two Ca^2+^ steps) were performed. They were then washed with Buffer B (200B) and elution with Buffer C was carried out according to Section 4.2.1 and resin regeneration and storage was conducted as in Section 4.3.1.

#### 4.3.4. FVIII and FIX Dynamic Binding Capacity and NaCl Elution on ANX Sepharose FF

The ANX Sepharose FF resin used was 51 mL and the used flow rates were as indicated in Section 4.2. 

Twenty CV of FFP were applied to the column and the unbound proteins were washed with 14 CV of Buffer A. FT was collected in fractions of 2 CV. 

Elution was carried out with a 5 CV linear gradient of 100 mM to 600 mM NaCl in Buffer A and 5 CV step of the Buffer A + 600mM NaCl. The elution was collected in fractions of 25 mL.

Resin regeneration and storage were performed as described in Section 4.3.1.

### 4.4. Chromatography of Plasma on DEAE Sepharose FF

The volume of DEAE Sepharose FF column was 28 mL and the used flow rates were as indicated in Section 4.2.

#### 4.4.1. DEAE Sepharose FF Chromatography without Ca^2+^ Gradient

Equilibration, sample, and washings with Buffers A and B were carried out according to Section 4.1, Section 4.2.1 and Section 4.4. Elution was carried out with 4 CV Buffer C. 

Resin regeneration and storage were performed as described in Section 4.2.1.

#### 4.4.2. DEAE Sepharose FF Chromatography with Ca^2+^ 5 mM to 50 mM Linear Gradient 

Equilibration, sample, and washing with 10 CV of Buffer A were performed according to Section 4.1 and Section 4.4. Next, elution with 10 CV linear gradient of 5 mM to 50 mM CaCl_2_ (in Buffer A) was performed. After the linear gradient, 5 CV Buffer A+ 50 mM CaCl_2_ was applied. Then a second wash with 5 CV Buffer A and a second elution with Buffer D (25 mM sodium citrate, NaCl 400 mM and 5 mM CaCl_2_, pH 6.0) was carried out. An additional wash with Buffer C was performed. Resin regeneration and storage were carried out as in Section 4.2.1.

### 4.5. Chromatography of Plasma on Fractogel EMD TMAE 

The flow rate was 40 mL min^−1^ during plasma application, while during other purification steps it was 60 mL min^−1^. The volume of the Fractogel EMD TMAE column was 28 mL 

#### 4.5.1. Fractogel EMD TMAE Chromatography without Ca^2+^ Gradient

Equilibration, sample, and washing with Buffers A were performed according to Section 4.1, Section 4.2.1 and Section 4.5. Elution was carried out with 4 CV Buffer C.

The resin was regenerated by sequentially washing with 2 CV 0.5M NaOH (with 1 h pause), 5 CV 2M NaCl and 10 CV purified water and stored in 10 mM NaOH.

#### 4.5.2. Fractogel EMD TMAE Chromatography with Ca^2+^ 5 mM to 50 mM Linear Gradient

Equilibration, sample, washing with 10 CV of Buffer A were performed according to Section 4.1 and Section 4.5. The following experimental conditions were carried out according to Section 4.4.2. Resin regeneration and storage followed the conditions described in Section 4.5.1.

### 4.6. Chromatography of Plasma on Fractogel EMD DEAE- Ca^2+^ 5 mM to 50 mM Linear Gradient 

The volume of the Fractogel EMD DEAE column was 24 mL and flow rates were as indicated in Section 4.5. 

The experimental procedure of this experiment was the same as in Section 4.5.2.

### 4.7. Analytical Methods

#### 4.7.1. Protein Quantification 

Total protein concentration was determined by the Bradford method [28], using bovine serum albumin (Pierce, Rockford, IL, USA) as standard.

#### 4.7.2. SDS-PAGE Analysis 

SDS-PAGE were performed on a 7.5% polyacrylamide gel under reducing (β-mercaptoethanol) and non-reducing conditions, on a 6% gel under non-reducing conditions, and on a 5 to 15% gradient under reducing conditions following Laemmli’s procedure [29].

#### 4.7.3. Determination of FVIII, Protein C, FIX and Prothrombin 

Activities of FVIII, Protein C, FIX and Prothrombin were evaluated using the chromogenic method in microplates, as recommended by the manufacturer. Samples of the FFP pool were used to build the calibration curve for each experiment. Considering that, by definition, one unit of FVIII is the amount of FVIII activity in one mL of normal plasma, the FVIII activity was expressed in arbitrary unit (U), which was used to calculate the percentage of the recovered activity and the specific activity. The purification factors were calculated as the ratio of the specific activity of the elution fraction by the specific activity of the loaded sample.

#### 4.7.4. Identification of the Proteins Eluted with CaCl_2_ by Mass Spectrometry

SDS-PAGE bands from the ANX Sepharose FF stepwise CaCl_2_ gradient purifications were excised for trypsin (Sigma-Aldrich, St. Louis, MO, USA) digestion and analyzed in a Autoflex Speed MALDI-TOF/TOF mass spectrometer (Bruker Daltonics, Billerica, MA, USA). Samples from non-reducing gels were treated with dithiotreitol and iodoacetamide and then digested by trypsin. The obtained peptides were applied to a reversed-phase-HPLC column. The collected fractions were analyzed.

##### Mass Spectra Acquisition and Peptide Sequencing 

Samples were prepared on MTP AnchorChips^TM^ 800/384 (Bruker Daltonics, Bilerica, MA, USA) by mixing 1 µL of each trypsin hydrolysate with 3 µL of matrix (2-cyano-3-(4-hydroxyphenyl)prop-2-enoic acid, α-CHCA, 10 mg/mL) in 1:1 (*v*/*v*) aqueous acetonitrile containing 0.3% TFA. The ionization was performed in the positive reflected mode by using the following instrument voltage parameters: Ion source 1: 20.00 kV, Ion source 2: 17.65 kV, Lens: 7.50 kV, Reflector: 22.00 kV, Reflector 2: 9.80 kV. Data were recorded in the m/z range from 600 to 4000 and analyzed by the Flex Analysis software (Bruker Daltonics, Bilerica, MA, USA). Peptide fragmentation was conducted by the LIFT™ methodology [30] with the following instrument voltage parameters: Ion source 1: 6.00 kV, Ion source 2: 5.25 kV, Lens: 3.00 kV, Reflector 1: 27.00 kV, Reflector 2: 11.80 kV, LIFT 1, 19.00 kV, LIFT 2: 4.70 kV. Assignment of the peaks corresponding to the y and b series was manually performed by using the Flex Analysis. Some preferential fragmentations of the peptide chain were considered [31]. Protein identification was performed by using the MS-Homology Search tool of the Protein Prospector package (http://prospector.ucsf.edu).

## Figures and Tables

**Figure 1 pharmaceuticals-15-01192-f001:**
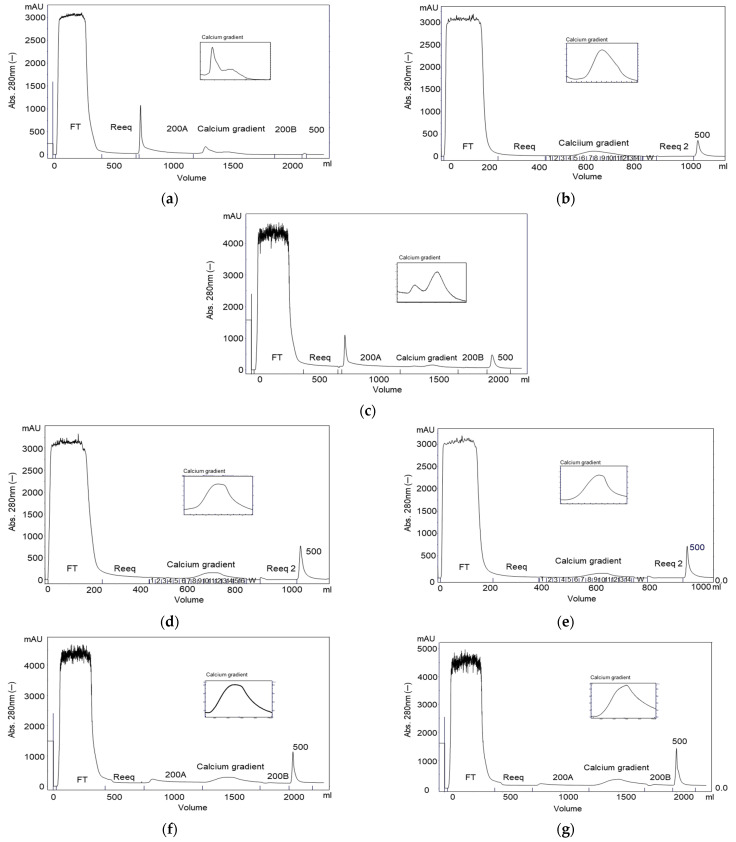
Chromatograms of the purifications of plasma on: (**a**) Q Sepharose FF with 5 mM to 100 mM linear CaCl_2_ gradient and citrate buffer; (**b**) Fractogel EMD TMAE with 5 to 50 mM linear CaCl_2_ gradient and citrate buffer; (**c**) ANX Sepharose FF, 5 to 25 mM linear CaCl_2_ gradient and citrate buffer; (**d**) DEAE Sepharose FF, 5 to 50 mM linear CaCl_2_ gradient and citrate buffer; (**e**) Fractogel EMD DEAE, 5 to 50 mM linear CaCl_2_ gradient and citrate buffer; (**f**) ANX Sepharose FF, 5 to 25 mM linear CaCl_2_ gradient and Bis-tris buffer; and (**g**) ANX Sepharose FF, 5 to 25 mM linear CaCl_2_ gradient and MES buffer.

**Figure 2 pharmaceuticals-15-01192-f002:**
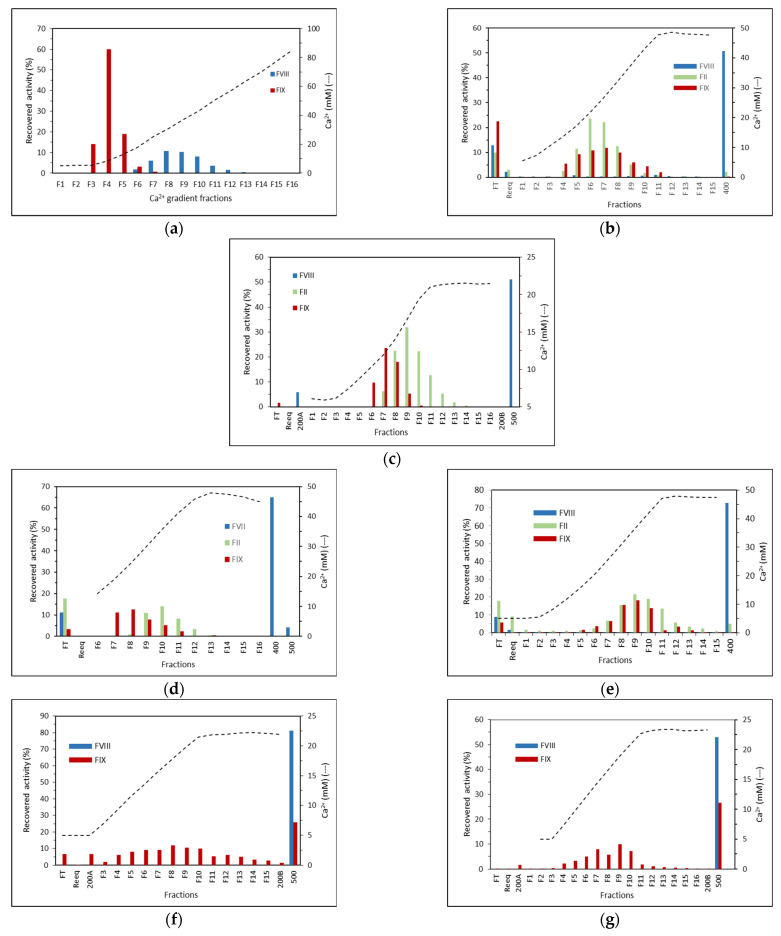
(**a**) FVIII and FIX recovered activities on the chromatography of plasma on Q Sepharose FF with 5 to 100 mM linear CaCl_2_ gradient and citrate buffer; (**b**) FVIII, FII and FIX recovered activities on the chromatography of plasma on Fractogel EMD TMAE with 5 to 50 mM linear CaCl_2_ gradient and citrate buffer; (**c**) FVIII, FII and FIX recovered activities on the chromatography of plasma on ANX Sepharose FF, 5 to 25 mM linear CaCl_2_ gradient and citrate buffer; (**d**) FVIII, FII and FIX recovered activities on the chromatography of plasma on DEAE Sepharose FF, 5 to 50 mM linear CaCl_2_ gradient and citrate buffer; (**e**) FVIII, FII and FIX recovered activities on the chromatography of plasma on Fractogel EMD DEAE with 5 to 50 mM linear CaCl_2_ gradient and citrate buffer; (**f**) FVIII and FIX recovered activities on the chromatography of plasma on ANX Sepharose FF, 5 to 25 mM linear CaCl_2_ gradient and Bis-tris buffer; and (**g**) FVIII and FIX recovered activities on the chromatography of plasma on ANX Sepharose FF, 5 to 25 mM linear CaCl_2_ gradient and MES buffer.

**Figure 3 pharmaceuticals-15-01192-f003:**
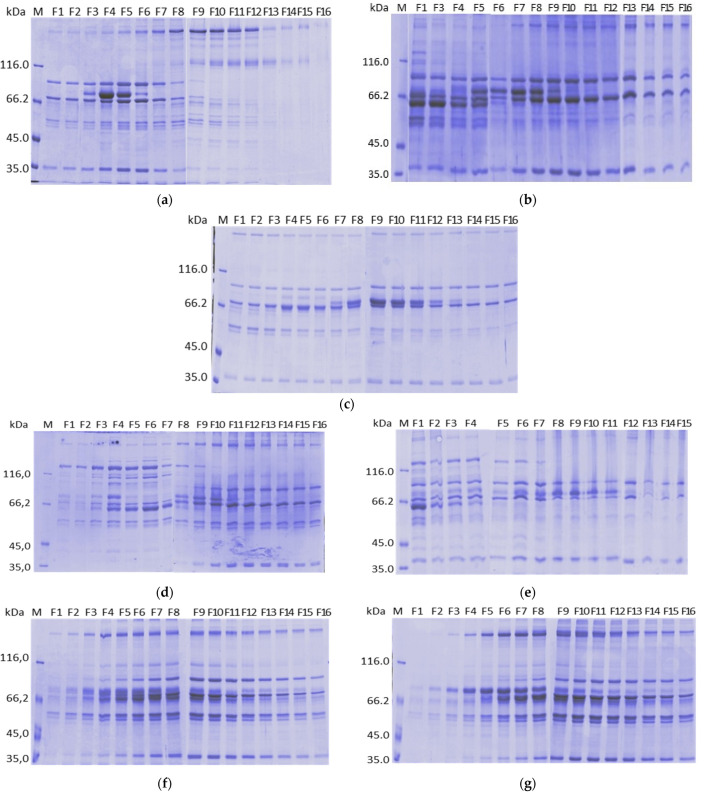
SDS-PAGE (7.5%) analysis under reducing conditions of purification of plasma on: (**a**) Q Sepharose FF with 5 to 100 mM linear CaCl_2_ gradient and citrate buffer; (**b**) Fractogel EMD TMAE with 5 to 50 mM linear CaCl_2_ gradient and citrate buffer; (**c**) ANX Sepharose FF with 5 to 25 mM linear CaCl_2_ gradient using citrate buffer; (**d**) DEAE Sepharose FF with 5 to 50 mM linear CaCl_2_ gradient and citrate buffer; (**e**) Fractogel EMD DEAE with 5 to 50 mM linear CaCl_2_ gradient and citrate buffer; (**f**) ANX Sepharose FF with 5 to 25 mM linear CaCl_2_ gradient and Bis-tris buffer; and (**g**) ANX Sepharose FF with 5 to 25 mM linear CaCl_2_ gradient and MES buffer. Lanes: M molecular mass marker; F1 to F16 fractions collected on linear CaCl_2_ gradient.

**Figure 4 pharmaceuticals-15-01192-f004:**
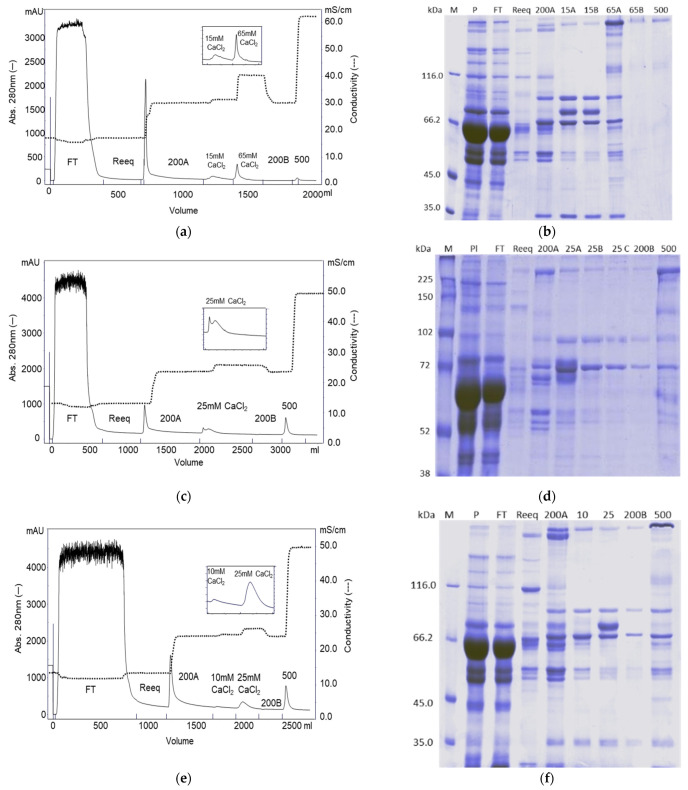
(**a**) Chromatogram of purification of plasma on Q Sepharose FF with stepwise 15 and 65 mM CaCl_2_ gradient; (**b**) SDS-PAGE 7.5% analysis under reducing conditions of purification of plasma on Q Sepharose FF with stepwise 15 and 65 mM CaCl_2_ gradient. Lanes: M- molecular mass marker; P- plasma; FT- flow through; Reeq- reequilibrium fraction; 200A- first wash with Buffer B;15A and 15B- first and second fractions eluted with 15 mM CaCl_2_ in 25 mM citrate, 200 mM NaCl, pH 6.0 buffer; 65A and 65B- fractions eluted with 65 m CaCl_2_ in 25 mM citrate, 200 mM NaCl, pH 6.0 buffer; 200B- second wash with buffer B; and 500- fraction eluted with Buffer C; (**c**) Chromatogram of purification of plasma on ANX Sepharose FF with stepwise 25 mM CaCl_2_ gradient; (**d**) SDS-PAGE 7.5% analysis under reducing conditions of purification of plasma on ANX Sepharose FF with stepwise 25 mM CaCl_2_. Lanes: M: molecular mass marker; P: plasma; FT: flowthrough; Reeq- reequilibrium fraction; 200A: first wash with Buffer B; 25A, 25B and 25C: first, second and third fractions eluted with 25 mM CaCl_2_ in 25 mM citrate, 200 mM NaCl, pH 6.0 buffer; 200B: second wash with buffer B; and 500- fraction eluted with Buffer C; (**e**) Chromatogram of purification of plasma on ANX Sepharose FF with stepwise 10 mM and 25 mM CaCl_2_ gradient; and (**f**) SDS-PAGE 7.5% analysis under reducing conditions of purification of plasma on ANX Sepharose FF with stepwise 10 mM and 25 mM CaCl_2_ gradient. Lanes: Lanes: M: molecular mass marker; P: plasma; FT: flowthrough; Reeq: reequilibrium fraction; 200A: first wash with Buffer B; 10: fractions eluted with 10 mM CaCl_2_ in 25 mM citrate, 200 mM NaCl, pH 6.0 buffer; 25: fractions eluted with 25 mM CaCl_2_ in 25 mM Citrate, 200 mM NaCl, pH 6.0 buffer; 200 B: second wash with Buffer B; and 500: fraction eluted with Buffer C.

**Figure 5 pharmaceuticals-15-01192-f005:**
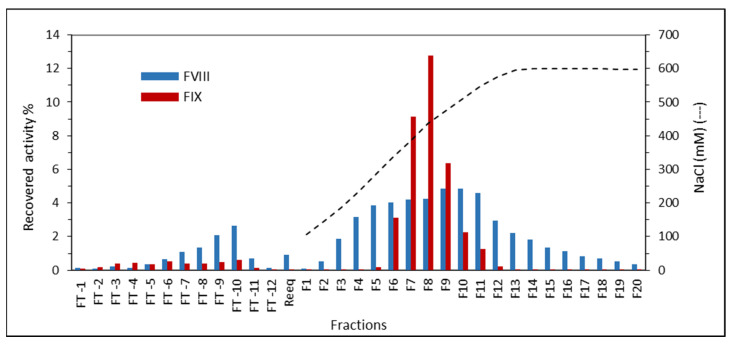
FVIII and FIX dynamic binding capacity and NaCl elution in purification of plasma on ANX Sepharose FF.

**Figure 6 pharmaceuticals-15-01192-f006:**
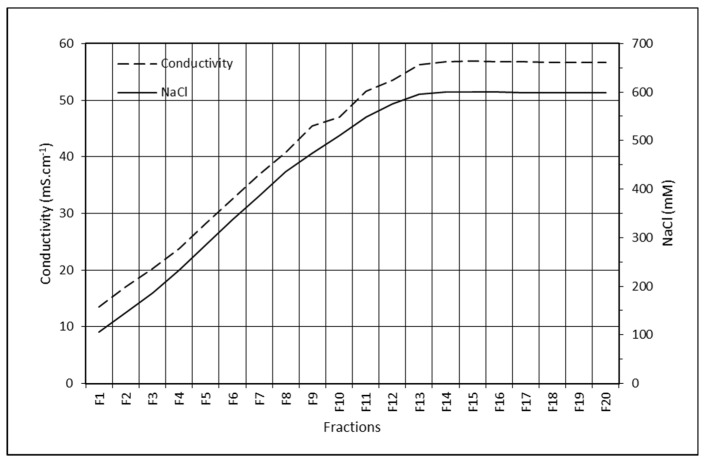
Relationship between Conductivity and NaCl concentration in 25 mM citrate, 5 mM CaCl_2_, pH 6.0 buffer.

**Figure 7 pharmaceuticals-15-01192-f007:**
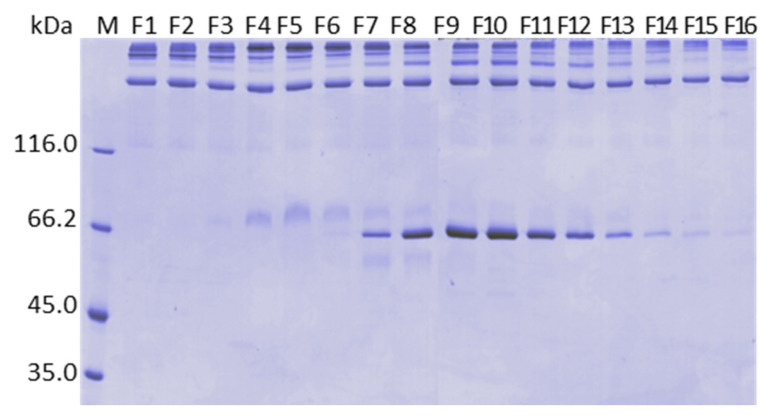
SDS-PAGE 7.5% analysis under non-reducing conditions of the purification of plasma on ANX Sepharose FF with CaCl_2_ 5 to 25 mM linear gradient in 25 mM citrate, 200 mM NaCl, pH 6.0 buffer.

**Figure 8 pharmaceuticals-15-01192-f008:**
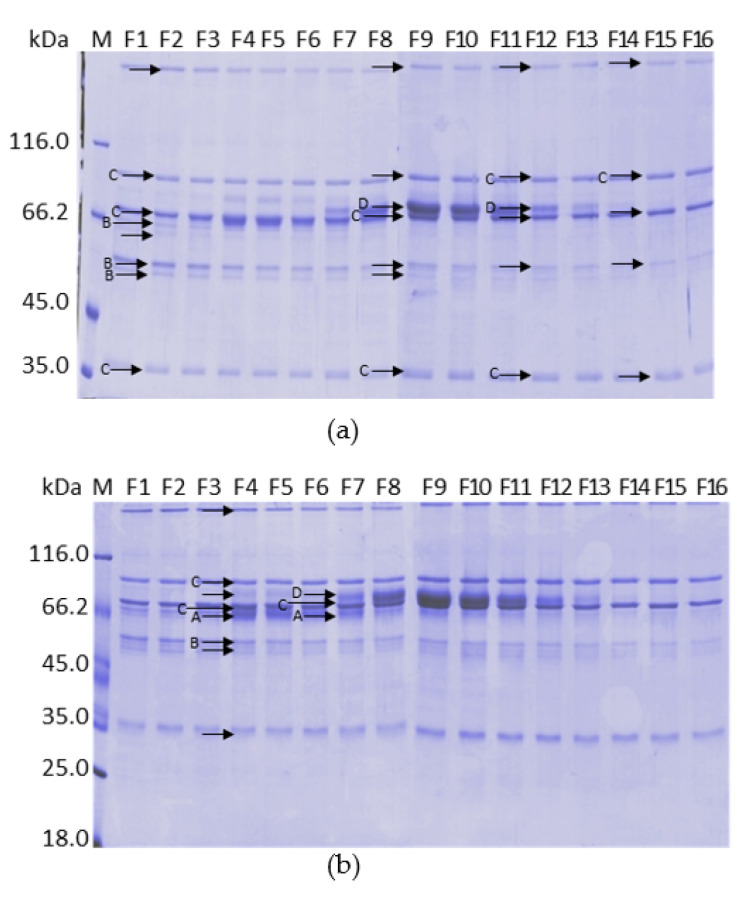
SDS-PAGE 7.5% (**a**) and 5% to 15% (**b**) analysis under reducing conditions of purification of plasma on ANX Sepharose FF with 5 to 25 mM linear CaCl_2_ gradient using citrate buffer. Bands marked with an arrow were analyzed by mass spectrometry. The identified proteins were: [A] Complement C4b Binding Protein (C4bBP); [B] Fibrinogen; [C] Complement C4; and [D] Prothrombin (FII).

**Figure 9 pharmaceuticals-15-01192-f009:**
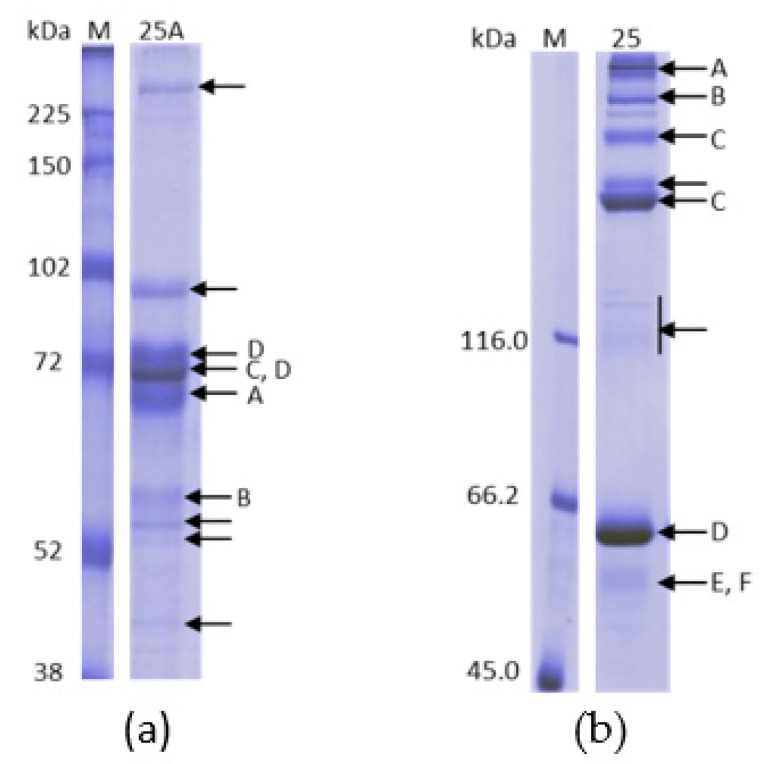
(**a**) SDS-PAGE 7.5% analysis under reducing conditions of fraction 25A of the purification of plasma on ANX Sepharose FF with stepwise 25 mM CaCl_2_ gradient. Bands marked with an arrow were analyzed by mass spectrometry. The identified were: [A] Complement C4b Binding Protein (C4bBP); [B] Fibrinogen; [C] Complement C4; and [D] Prothrombin (FII); and (**b**) SDS-PAGE 6% analysis under non-reducing conditions of fraction 25 of purification of plasma on ANX Sepharose FF with stepwise 10 mM and 25 mM CaCl_2_ gradient. Bands marked with an arrow were analyzed by mass spectrometry. The identified were: [A] Complement C4b Binding Protein (C4bBP); [B] Fibrinogen; [C] Complement C4; [D] Prothrombin (FII); [E] Factor IX (FIX); and [F] Protein Z.

**Table 1 pharmaceuticals-15-01192-t001:** FVIII purifications from plasma.

Experiment		Specific Activity in Plasma (U mg^−1^) ^#^	FVIII Recovered Activity from Plasma (%)	Specific Activity in the Eluate (U mg^−1^)	Purification Factor
Resin Ca^2+^ Gradient (mM)	Buffer				
Q Sepharose FF 15 and 65 *	citrate	1.8	60	360	201
ANX Sepharose FF 5–25	citrate	1.7	51	271	157
DEAE Sepharose FF 5–50	citrate	1.2	65	162	130
Fractogel EMD TMAE 5–50	citrate	2.3	51	327	143
Fractogel EMD DEAE 5–50	citrate	2.5	73	368	147
ANX Sepharose FF 5–25	Bis-tris	1.5	67	309	211
ANX Sepharose FF 5–25	MES	1.4	53	178	125
ANX Sepharose FF 25	citrate	1.7	68	433	250
ANX Sepharose FF 10 and 25	citrate	1.0	59	228	220

^#^ The specific activity was calculated by dividing the activity per mL (U mL^−1^) by the protein concentration in mg per mL (mg mL^−1^). * FVIII activity in the 65 mM CaCl_2_ fraction.

**Table 2 pharmaceuticals-15-01192-t002:** Proteins identified by mass spectrometry of fractions collected from plasma purification on ANX Sepharose FF using linear 5 to 25 mM CaCl_2_ gradient.

Protein	Fragment	Obtained Mass (m/z)	Calculated Mass (m/z)	Protein Mass (kDa)	Identification on Gels
C4bBP	(K)LSLEIEQLELQR	1470.9	1470.8	570	[A]
Fibrinogen	(K)GLIDEVNQDFTNR	1520.7	1520.7	340	[B]
	(K)IHLISTQSAIPYALR	1683.1	1683.0		
	(K)HQLYIDETVNSNIPTNLR	2127.1	2127.1		
Complement C4	(R)SFFPENWLWR	1381.7	1381.7	202	[C]
	(R)TTNIQGINLLFSSR	1563.8	1563.8		
	(R)YIYGKPVQGVAYVR	1612.8	1612.9		
	(K)VGLSGmAIADVTLLSGFHALR	2144.4	2144.2		
	(R)ALEILQEEDLIDEDDIPVR	2225.0	2225.1		
	(R)VTASDPLDTLGSEGALSPGGVASLLR	2483.2	2483.3		
	(R)LTVAAPPSGGPGFLSIERPDSRPPR	2574.3	2574.4		
	(R)YVSHFETEGPHVLLYFDSVPTSR	2680.4	2680.3		
	(R)STQDTVIALDALSAYWIASHTTEER	2778.2	2778.4		
	(R)EC(Propionamide)VGFEAVQEVPVGLVQPASATLYDYYNPER	3514.8	3514.7		
Prothrombin (FII)	(K)YGFYTHVFR	1189.6	1189.6	70	[D]
	(R)TATSEYQTFFNPR	1561.8	1561.7		

C (Propionamide) indicates a cysteine residue modified by acrylamide, m is a methionine S-oxide.

**Table 3 pharmaceuticals-15-01192-t003:** Proteins identified by mass spectrometry of the fraction “25A” from plasma purification on ANX Sepharose FF using stepwise 25 mM CaCl_2_ gradient.

Protein	Fragment	Obtained Mass (m/z)	Calculated Mass (m/z)	Protein Mass (kDa)	Identification on Gels
C4bBP	(K)EDVYVVGTVLR	1249.7	1249.7	570	[A]
	(K)LSLEIEQLELQR	1470.9	1470.8		
	(R)KPDVSHGEMVSGFGPIYNYK	2241.1	2241.1		
Fibrinogen	(R)HQLYIDETVNSNIPTNLR	2127.3	2127.1	340	[B]
Complement C4	(K)FACYYPR	990.5	990.4	202	[C]
	(R)GLQDEDGYR	1052.6	1052.5		
	(A)SATLYDYYNPER	1491.8	1491.7		
	(R)YIYGKPVQGVAYVR	1613.0	1612.9		
	(R)GHLFLQTDQPIYNPGQR	1984.6	1984.0		
	(R)LTVAAPPSGGPGFLSIERPDSRPPR	2574.6	2574.4		
	(R)YVSHFETEGPHVLLYFDSVPTSR	2680.5	2680.3		
Prothrombin (FII)	(K)YGFYTHVFR	1189.7	1189.6	70	[D]
	(R)TATSEYQTFFNPR	1561.9	1561.7		
	(K)LAACLEGNCAEGLGTNYR	1854.8	1854.8		

Cysteine residues were either modified by acrylamide, C(Propionamide) or were unmodified, as for the peptide with m/z 1854.8.

**Table 4 pharmaceuticals-15-01192-t004:** Proteins identified by mass spectrometry of the fraction “25” from plasma purification on ANX Sepharose FF using stepwise 10 mM and 25 mM CaCl_2_ gradient.

Protein	Fragment	Obtained Mass (m/z)	Calculated Mass (m/z)	Protein Mass (kDa)	Identification on Gels
C4bBP	(R)FSAIC(CAM)QGDGTWSPR	1581.8	1581.7	570	[A]
	(R)GVGWSHPLPQC(CAM)EIVK	1707.0	1706.9		
Fibrinogen	(K)VQHIQLLQK	1106.9	1106.6	340	[B]
	(K)YEASLITHDSSIR	1492.0	1491.7		
	(T)ADSGEGDFLAEGGGVR	1536.0	1536.7		
Complement C4	(R)SFFPENWLWR	1381.7	1381.7	202	[C]
	(R)YIYGKPVQGVAYVR	1612.9	1612.9		
	(R)ALEILQEEDLIDEDDIPVR	2225.2	2225.1		
Prothrombin (FII)	(K)YGFYTHVFR	1189.6	1189.6	70	[D]
	(R)TATSEYQTFFNPR	1561.8	1561.7		
	(K)GQPSVLQVVNLPIVERPVC(CAM)K	2232.3	2232.2		
	(K)HQDFNSAVQLVENFC(CAM)R	1963.9	1963.9		
	(R)SEGSSVNLSPPLEQC(CAM)VPDR	2071.0	2071.0		
	(R)ITDNmFC(CAM)AGYKPDEGKR	2018.0	2017.9		
Factor IX	(K)FGSGYVSGWGR	1172.7	1172.5	56	[E]
	(K)SCEPAVPFPC(CAM)GR	1376.8	1376.6		
Protein Z	(K)DFAEHLLIPR	1210.9	1210.7	62	[F]

C(CAM) indicates the cysteine residues modified by carbamidomethylation with iodoacetamide, m is a methionine *S*-oxide.

## Data Availability

Data is contained within the article.

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
