# Peer review of "Non-Cryoprecipitation Separation of Coagulation FVIII and Prothrombin Complex Proteins by Pseudoaffinity Calcium Elution Chromatography Using Anion Exchange Resin"

_pharmaceuticals, 2022, doi:10.3390/ph15101192_

Round 1
Reviewer 1 Report
The manuscript by Feliciano et al. described the development and basic characterization of a method to separate coagulation factor VIII from prothrombin complex concentrate on anion exchange resins by subjecting plasma to a Ca2+ gradient by pseudoaffinity chromatography first before NaCl elution. The separation of the two major fractions is analogous to the conventional cryoprecipitate (FVIII-rich) vs cryosupernatant (FIX, FII & FX-rich) fractions, but can be performed on a single anion exchange matrix without the need for low temperature. Although the need for further validations such as purity and stability analyses is obvious, I view this as a comprehensive proof-of-concept account for a new process and has merit for publication. Overall, this is an excellent study, covered five different resins with good variety in matrix, chemistry, buffers, and elution conditions tested. Unfortunately, generalization of rules governing the elution is not apparent from the data presented.
I have no major issues with this manuscript other than the following comments:
1. Overall, the writing should also be more concise. I find there are quite a lot of overlaps between results & methods (e.g. 1st paragraph of Results reads like Methods), as well as between results & discussion.
2. Since the nature of this study is to test multiple different conditions with minor changes between each, sometimes it can be confusing when results of from the same resin (chromatogram vs activity vs SDS-PAGE in Fig. 1-3) have to be compared over several figures. As the text are presenting one set of data from essential the same condition (eg. Section 2.1), I find myself having to flip through several pages just to follow the train of thought as presented by the authors (i.e. the flow of figures goes from 1a to 2a to 3a, plus 4a & 4b for step-wise gradients). Personally, I feel this way of data presentation reduced readability of the manuscript. Some of my suggestion would be to place Fig. 1a, 2a, & 3a side-by-side in the same figure for the ease of comparisons. Perhaps the comparisons between buffers (Bis-Tris vs MES vs citrate) in ANX Sepharose FF should be the first set of data presented to establish the superiority of citrate buffer, since all subsequent conditions are focused on citrate alone. Other rearrangements are also welcomed, as long as readability is improved.
3. For processes, reproducibility is obviously important. I would appreciate if the authors can include descriptions of the number of independent experiments with possibly the inclusion of estimation of errors (for example, but not limited to, Table 1).
Reviewer 2 Report
The study by Feliciano et al. evaluates a new separation technique for coagulation factor VIII and PCC. The authors should be complimented for their efforts in investigating this technique, as it may reduce the costs of purification of coagulation factors. Ultimately, making these products available not only for genetic disorders but also for acquired bleeding disorders.
I do have some minor points that could be addressed to improve their paper:
Minor:
Because you go into many different points (hemophilia, recombinant techniques, the use of plasma over the world, the roles of different coagulation factors, the current purification technique) in your introduction the focus is somewhat lacking. I would suggest focusing on your introduction and coming to your point on the separation technique earlier. I must go to 2/3 of your introduction to arrive at your research topic.
Introduction
I miss a clear hypothesis of your research? What did you expect?
Results
Overall a well-presented result section!
However, some of the methods are described in your results:
Line 105- 123 – this whole section should be in your methods. Some parts can be results, but mention them as results. Please do not mix methods in your result section.
Just out of curiosity did you use AB plasma? And what was the time from collection to thawing?
Discussion
Here I echo my thoughts on focus. Some parts are more results sections. What are the main results you have found and how does this relate to the literature? You don’t have to repeat the results you have just shown your reader. More important is the implication. Also, you introduce the complement system while I think most readers will know what complement is, but I still am not sure why it is important that you identified C4BP and C4 in this technique. What is the implication of this result?
Also I am not sure but the format of discussion – methods – conclusion is not how I would design my paper. Either introduction – results – discussion/conclusion – methods or introduction – methods – results – discussion/conclusion. The structure now makes people miss your conclusion section
Your conclusion/end of your discussion does not resonate well with your aim in your introduction. Also, I would like to know what future investigations should be focused on.
Overall, this is a well-conducted study that needs some rewriting to clarify your message.
